# Development and Effectiveness of a Rapid Cycle Deliberate Practice Neonatal Resuscitation Simulation Program: A Quasi-Experimental Study

**DOI:** 10.3390/healthcare12010104

**Published:** 2024-01-02

**Authors:** Sun-Yi Yang, Yun-Hee Oh

**Affiliations:** 1College of Nursing, Konyang University, Daejeon Medical Campus, 158, Gwanjeodong-ro, Seo-gu, Daejeon 35365, Republic of Korea; 2Department of Nursing, Cheju Halla University, 38, Halladaehak-ro, Jeju-si 63092, Republic of Korea; Allegra600@hanmail.net

**Keywords:** rapid cycle deliberate practice, high-fidelity simulation training, students, nursing, intensive care units, neonatal

## Abstract

The Rapid Cycle Deliberate Practice (RCDP) simulation during neonatal resuscitation program (NRP) training provides in-event feedback for each simulation step, repeats the simulation from the beginning, and undergoes a continuous improvement process. It also offers after-event debriefing that involves follow-up discussion and reflection after completing simulations. These two methods differ in the timing and frequency of feedback application, and there may be differences in the effectiveness of neonatal resuscitation training. A quasi-experimental simulation study with a pre- and post-test design was used; the experimental group received RCDP simulation NRP training, based on the self-determination theory, while the control group received an after-event debriefing, following the NRP scenario. The experimental group displayed significantly improved clinical decision-making skills compared with the control group. When responding to emergencies involving high-risk newborns, we found that RCDP simulation during NRP training and better preparation for neonatal resuscitation among nursing students improved outcomes for newborns.

## 1. Introduction

Since the novel coronavirus (COVID-19) pandemic started in 2020, certain clinical practicums were suspended for the safety of both patients and nursing students [1,2]. In particular, neonatal intensive care unit (NICU) practicums were unlikely to resume because of the high risk for infection among immunocompromised newborns [2].

High-risk neonatal simulation training methods have been attracting public attention as a means of overcoming the restrictions of limited NICU in-event training [3]. Simulation training has the advantage of utilizing high-fidelity simulators in a practice environment that closely mimics a clinical environment, thus creating situations that achieve the intended learning objectives and provide repeated hands-on practice and feedback to nursing students [4]. 

Simulation training comprises a pre-briefing, the scenario, and a debriefing; debriefing is a crucial step in improving learning effectiveness, and the application of after-event debriefing has been emphasized by previous studies [5,6]. An after-event debriefing involves a follow-up discussion and evaluation and reflection after completing the simulation run; although it utilizes an advocacy debriefing approach, it offers less of an opportunity to correct skills and practice [7,8].

Rapid Cycle Deliberate Practice (RCDP), however, was developed for emergency clinical training at Johns Hopkins University; this mastery learning method uses within-event debriefing. RCDP utilizes a training method that pauses the simulation when a learner’s error is discovered during the scenario and immediately provides directive and personalized evidence-based feedback (solution-oriented debriefing, directing appropriate methods and techniques) for each step; thereafter, the simulation restarts the scenario from the beginning, and the problem situation is progressively expanded [9,10]. RCDP is repetitive training, and, therefore, offers a continuous improvement process, as well as the advantage of enabling the trainee to correct mistakes and promoting muscle and long-term memory through continuous practice [8]. RCDP has been associated with improved team performance, as it encourages trainees to take action as quickly and accurately as possible, which saves lives and prevents negative after-effects [11]. The use of the RCDP in-event feedback in resuscitation training helped achieve the goal of mastery learning by familiarizing students with resuscitation guidelines and enhancing their skills and competency [10]. Furthermore, Blanchard et al. [9] found RCDP to be suitable for training algorithmic, time-sensitive, team-based procedural interventions. Some studies provided RCDP neonatal resuscitation training to experts (physicians) and demonstrated significant effectiveness; however, these previous studies focused mainly on resuscitation techniques and skill evaluation, and it was difficult to find studies that included intervention strategies to improve the decision-making ability for each stage of resuscitation and evaluate their effectiveness [9,12,13]. Moreover, previous studies comparing the effectiveness of NRP interventions using the RCDP in-event feedback with after-event debriefing were focused on physicians or inter-professional education involving physicians and registered nurses, and studies targeting beginners (nursing students) were difficult to find [8,14]. Furthermore, no previous studies have seemingly applied theoretical frameworks to RCDP training as an intervention strategy.

This study utilizes Deci and Ryan’s [15] self-determination theory, a model developed to promote human motivation and healthy psychological and behavioral functioning. The self-determination theory has been used effectively in various motivational training programs [16,17]. Therefore, a systematic review by Taras and Everett [10] argued that if RCDP training were to use the self-determination theory to provide feedback and opportunities to repeat tasks, learners would become more competent. 

This study aimed to develop and apply the RCDP neonatal resuscitation program (NRP), based on Deci and Ryan’s [18] self-determination theory model, to improve nursing students’ skills and to validate its effectiveness in a simulated environment by comparing it with traditional instructional methods (Figure 1). The outcome variables for effectiveness validation included NRP knowledge, self-confidence, and competence, as well as clinical decision-making skills, simulation effectiveness, and debriefing satisfaction, which were utilized in previous studies [9,19,20,21].

**H1.** 
*NRP knowledge will increase more in the experimental group applying NRP training utilizing RCDP in-event training than in the control group, utilizing after-event debriefing and a traditional debriefing method.*


**H2.** 
*NRP students’ self-confidence will increase more in the experimental group than in the control group.*


**H3.** 
*Students’ clinical decision-making skills will increase more in the experimental group than in the control group.*


**H4.** 
*Students’ simulation effectiveness will increase more in the experimental group than in the control group.*


## 2. Materials and Methods

### 2.1. Research Design

This study was designed as a quasi-experimental study with a pre- and post-test design to apply RCDP neonatal resuscitation simulation training (Figure 2). 

### 2.2. Participants

This study was conducted at J City University, with the permission of the head of the department, after explaining the nature and purpose of the study. The target population comprised senior nursing students from the university’s bachelor’s degree program. The selection criterion was that all participants had to be prelicensure nursing students; those who were absent during the semester, who had more than one year of experience studying or working before entering the nursing program, or who had previously participated in NRP simulation-related programs in addition to their coursework were excluded, as these conditions were deemed likely to affect the study results. The experimental or control group was assigned by convenience sampling. Three to four participants from the same group were assigned to participate in each scenario as a team. 

The number of participants was estimated using G^*^Power version 3.1.9.7 [22]. For the hypothesis test, the determined minimum sample size per group was 26 participants, for a total of 52, which was calculated using a power of 0.80, significance level of α = 0.05, and effect size (large) of d = 0.80 for the comparison of the two group means. Previous studies on the effectiveness of RCDP reported dropout rates ranging from 0% to 6.1% [23,24,25]. The initial sample size of this study was 29 participants per group (58 participants), and the estimated dropout rate was 10%. One participant in the control group was excluded from the study for non-response. Finally, the experimental group had 29 participants (100.0%), and the control group had 28 participants (96.6%) (*N* = 57). 

### 2.3. Study Protocol

The instructor during the experiment was an independent third party with 17 years of experience as an NICU nurse and more than 10 years of experience teaching high-risk neonatal simulation classes; they received RCDP training from the Society for Simulation in Healthcare.

In week 1, all study participants received a 30-min lecture and a 70-min instructor-led skills demonstration on the eighth edition of the NRP guidelines, developed by the American Heart Association and the American Academy of Pediatrics [21].

In week 2, all participants, from both the experimental and control groups, were given a 30-min pre-briefing (Laerdal^®^ premature Anne, which is a high-fidelity simulator, introduction) and scenario briefing (non-personalized real-life scenario of a premature infant born at 24 weeks with difficulty breathing due to a premature rupture of the membrane and meconium stain), as well as a 70-min group (3–4 people) NRP laboratory. 

In week 3, the experimental group was given NRP simulation training with RCDP in-event feedback (5 × 20-min rounds = 100 min) (Appendix A), and the control group was given the traditional high-fidelity NRP simulation training. Following training, the control group received the after-event debriefing, Promoting Excellence and Reflective Learning in Simulation (100 min) (Appendix B) (Figure 2). 

### 2.4. Data Collection

Research approval was obtained from the Institutional Review Board of K University (IRB No. KYU-2022-01-011) after explaining the purpose, methods, rights of participants, and questionnaire items. All participants provided written informed consent before the start of the study. Participation in the study was voluntary, and gifticons were provided to all study participants.

The pre- and post-surveys were conducted from September to December 2022 using an online questionnaire to measure NRP knowledge, NRP self-confidence, and NRP competence, as well as clinical decision-making skills, simulation effectiveness, and debriefing satisfaction (20 min).

### 2.5. Research Tools

#### 2.5.1. NRP Knowledge

NRP knowledge was assessed using two measures: (1) Yoo’s [25] NRP knowledge scale, based on the core content of the American Heart Association’s NRP [26], and (2) the NRP^TM^, which reflects the eighth edition of the NRP from the American Academy of Pediatrics and the American Heart Association [21,27]. 

The NRP knowledge scale comprises 27 questions: The scores range from 0 to 27, with higher scores indicating greater NRP knowledge. This scale is standardized through reliability and validity testing, where Cronbach’s α was 0.87 when the instrument was developed and 0.72 in this study.

The NRP^TM^ was modified and supplemented for this study before being validated for content validity by a NICU head nurse and two professors of pediatric nursing. 

#### 2.5.2. NRP Self-Confidence

Similar to NRP knowledge, NRP self-confidence was measured using Yoo’s [25] NRP knowledge scale and the NRP^TM^ [27]. The confidence in caring for newborns with a respiratory distress syndrome instrument comprises 14 questions, measured using a five-point Likert scale. The scores range from 14 to 70, with higher scores indicating higher NRP self-confidence. Cronbach’s α was 0.96 when the instrument was developed and 0.98 in this study.

#### 2.5.3. Clinical Decision-Making

Clinical decision-making skills were measured using the “Clinical decision-making skill scale”, translated and adapted by Baek [28] from the “Clinical Decision-Making in Nursing Scale” by Jenkins [29]. The items are measured using a five-point Likert scale; scores range from 40 to 200, with higher scores indicating better clinical decision-making skills. Cronbach’s α was 0.77 when the instrument was developed and 0.87 in this study.

#### 2.5.4. Simulation Effectiveness

The simulation effectiveness factor was measured using the “Simulation Effectiveness Tool-Modified (SET-M)”, developed by METI^®^ [30]. Items were measured using a 3-point Likert scale; the scores range from 19 to 57, with higher scores indicating higher simulation effectiveness. Cronbach’s α was 0.83–0.94 when the instrument was developed and 0.97 in this study.

### 2.6. Data Analysis

Data were analyzed using IBM SPSS Statistics 27.0; the mean of the sample was analyzed by conducting the Shapiro–Wilke test, which was found to follow a normal distribution; therefore, a parametric statistical analysis method was used. The preliminary homogeneity of the general characteristics and dependent variables (NRP knowledge, self-confidence, and clinical decision-making skill) between the experimental group and control group was analyzed by conducting a chi-squared test, a Fisher’s exact test, and an independent *t*-test. The pre–post mean differences of the two groups measuring NRP knowledge, NRP self-confidence, clinical decision-making skill, simulation effectiveness, and debriefing satisfaction were analyzed by conducting an independent *t*-test. 

## 3. Results

### 3.1. Validating the Pre-Homogeneity of Participants’ General Characteristics

In the homogeneity test of participants’ general characteristics, the two groups were significantly different in terms of sex, nursing major satisfaction, clinical practice satisfaction, simulation class satisfaction, and desire to practice in a neonatal resuscitation simulation lab; thus, the homogeneity of the two groups was confirmed (Table 1). The preliminary homogeneity test for the dependent variables confirmed that the two groups were homogeneous (Table 1). 

### 3.2. Validating the Effectiveness of RCDP Simulation Based on the Self-Determination Theory Model

Table 2 summarizes the statistical analysis results. The post-hoc mean differences between the experimental and control groups were significant in terms of NRP knowledge (t[55] = 2.25, *p* = 0.028), NRP self-confidence (t[55] = 4.67, *p* < 0.001), clinical decision-making skill (t[55] = 6.03, *p* < 0.001), and simulation effectiveness (t[55] = 3.51, *p* = 0.001).

The analysis of the pre–post mean difference between the experimental and control groups revealed significant differences in NRP self-confidence (t[55] = 1.45, *p* = 0.045), clinical decision-making skills (t[55] = 3.13, *p* = 0.003), and simulation effectiveness (t[55] = 1.99, *p* = 0.042). 

## 4. Discussion

This study is significant in that it utilized eight strategies (choice, explanation/rationale, acknowledgment of feelings, optimal challenge, positive performance, caring, inclusive environment, and security) according to the three components (autonomy, competence, and relatedness) described in Deci and Ryan’s [18] self-determination theory model and verified its effectiveness.

The NRP knowledge of both the experimental and control groups improved significantly after the intervention, compared with before the intervention; however, there were no significant pre–post mean differences. In a previous study on RCDP and traditional debriefing methods to advance cardiac life support training among paramedic students, a significant improvement in knowledge after the intervention was also reported, thus supporting the results of this study [31]; however, the pre–post mean difference between the two groups was not presented, so it could not be compared. In this study, both groups were provided with online neonatal resuscitation pre-study and on-campus laboratory skill practice before the simulation run, which may have contributed to these results. However, existing research on medical residents found that subsequent reflective debriefing improved knowledge significantly more than RCDP training [9]. It would be worthwhile to determine whether these results are due to differences in the interventions or whether the participants were novices or experts. However, as both groups were allowed to practice their neonatal resuscitation skills in the classroom and laboratory in a pre-study, it would be unreasonable to conclude that the provision of feedback and debriefing alone improved their knowledge. Therefore, future studies must analyze the effect of providing a pre-study before the simulation on knowledge improvement. Furthermore, the long-term effect of knowledge must be analyzed to determine whether the effect of the intervention is sustained.

NRP self-confidence in neonatal nursing knowledge also significantly improved in both groups after the intervention, compared with before the intervention, as is evident in the pre–post mean difference. Previous studies [12,25,32] have also found significant improvements in students’ self-confidence with the RCDP training, which supports the findings of this study. This may be due to the nature of the RCDP method, which provides learners with a personalized and immediate correction of poorly performed skills, and opportunities to apply skills in iterative cycles so that, by the end of the training, they can be fully proficient. In particular, this study adopted a competency enhancement strategy of the self-determination theory model that provided step-by-step scenarios to perform optimal challenge tasks by sequentially increasing the difficulty level and correcting wrong skills after each round; this provided an opportunity for positive performance in the next round. In this study, we reinforced optimal challenges and positive performance as intervention strategies to promote competency, which is one of the components of the self-determination theory. This appears to have contributed to improving self-confidence in neonatal resuscitation. However, another study found that students who received after-event debriefing for emergency cardiovascular care were more confident than those who received RCDP feedback [9]. This may have been partially due to those in the RCDP group, second-year residents, constantly receiving negative feedback from experts in front of their peers, thus making them feel nervous and intimidated; this may have negatively affected their performance. This notion is supported by a previous study by Gantt et al. [33], which showed that there are differences in students’ preferences for debriefing, depending on their proficiency level; therefore, it is necessary to consider participants’ characteristics and the training purpose to determine the most effective debriefing method. Furthermore, it is important to ensure that students’ privacy is protected. One way of achieving this is allowing positive feedback to be shared in the presence of other learners, while negative feedback should be in the form of individualized notes that could only be received by the learner. Alternatively, compared with neonatal resuscitation, which has a set algorithm, it could be that RCDP feedback has limitations in improving confidence in simulation situations, in which there are many variables for each situation. In future research, the effectiveness of in-event feedback and after-event debriefing should be compared and verified, depending on the presence or absence of a nursing skill algorithm, and novices and experts should be compared.

The clinical decision-making skills of the experimental group significantly improved after the intervention, compared with before the intervention, as evident in the pre–post mean difference; however, this was not the case for the control group. Blanchard et al.’s [9] work supports our findings, demonstrating that the RCDP in-event feedback resulted in more accurate judgments of when to use a defibrillator and was a faster application than the traditional instructional method. In this study, participants were provided with the opportunity to voluntarily select their roles in a scenario to improve autonomy, a component of self-determination theory. In addition, providing an opportunity to make the right decisions during the scenario process by providing immediate feedback and explanation at the end of each round—in the case of an incorrect judgment—seems to have improved the learners’ clinical decision-making skills. However, there are certain limitations to concluding that RCDP training can be generalized to all situations, as it is appropriate to use in time-sensitive, algorithmic, clinical situations.

The simulation effectiveness of the experimental group improved significantly after the intervention, compared with the control group, as evident in the pre–post mean difference. In validating the effectiveness of the RCDP for training skills that must be performed in a timed manner, previous studies showed that it significantly improved skill performance, compared with the traditional instruction method, thus supporting the results of this study [11,12,34]. Furthermore, a study on NRP training for interns revealed that the time taken to implement a critical intervention was significantly reduced in the group that received RCDP training [12]. A systematic review by Taras and Everett [10] argued that RCDP training reflected the concept of intentional practice—a way to help learners achieve mastery through the intentional and repeated practice of a skill. In this study, to improve the relatedness of team members, learners were encouraged to perform each round at each stage. Teamwork was also encouraged while performing neonatal resuscitation. Training was conducted in a safe simulation room with settings as similar as possible to an NICU so that learners could immerse themselves in the training without being buried in their own mistakes. It seems that doing so contributed to producing this result. However, it is questionable whether performance in different cases would be similar to the results of this study because of the nature of RCDP training, which focuses on developing quick and accurate skills in simulated situations and does not allow for in-depth reflection afterward. As RCDP training can be meaningfully utilized as a refresher for intervention delivery or in real-world clinical settings, it would be beneficial to consider the introduction of a hybrid training approach that combines RCDP in-event feedback with the traditional after-event debriefing [12].

## 5. Conclusions

RCDP in-event feedback, which provides direct, personalized, and evidence-based feedback during the intervention process, rather than the traditional practice of debriefing after the intervention, resulted in significant effects on all outcome variables after the intervention, compared with before the intervention. After comparing the control group, who received the traditional instructional method and the subsequent reflective debriefing, and the experimental group, who received RCDP training, the latter was found to be more effective in improving all the outcome variables (NRP self-confidence, clinical decision-making skill, and simulation effectiveness) except NRP knowledge, which showed no significant difference between the two groups. The results suggest that the applicability of RCDP training to neonatal resuscitation training for novice nursing students was verified. As the repeated application of the simulation by each learner is limited because of the lack of high-fidelity simulators and facilitators relative to the number of learners, the RCDP training method seems to be an option to provide opportunities for complete learning by allowing learners to iteratively refine and perfect the intervention during a single simulation experience.

### Limitations

The limitations of this study and future research suggestions are as follows: First, the program’s effectiveness was assessed solely through pre-intervention and post-intervention measurements, and its sustained effectiveness was not verified through periodic retention evaluations. Therefore, future studies should follow up on the long-term memory consolidation of RCDP training. Second, with respect to validating the effectiveness of a short-term intervention, whether a study used a scale that can sensitively measure changes in the measured variable should be considered. It is suggested that qualitative analysis should be added to analyze students’ opinions on which elements of the course were valuable in enhancing their competencies and which courses they believed reduced their cognitive load; Third, there is no standardized approach to an RCDP method with directive, customized, and evidence-based feedback. As this can significantly influence the effectiveness of the competency of the feedback provider on training, standardized feedback delivery methods must be explored first. Furthermore, as it is important to secure the expertise and debriefing competence of the feedback providers, the competence of the feedback providers and its effect on training should be measured objectively. It is also necessary to develop a standardized approach for training RCDP educators.

## Figures and Tables

**Figure 1 healthcare-12-00104-f001:**
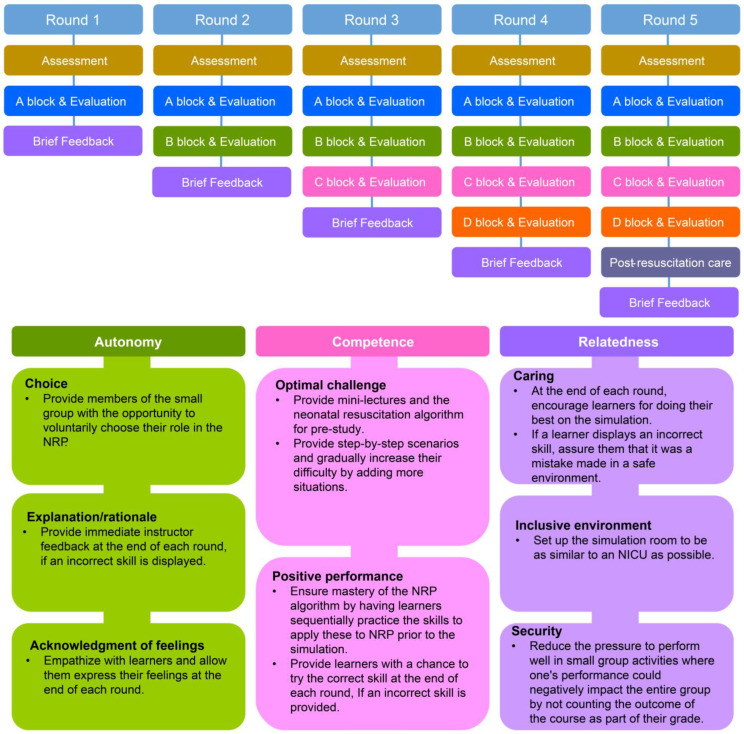
Study design and RCDP stepwise sequence and intervention strategies of the NRP program.

**Figure 2 healthcare-12-00104-f002:**
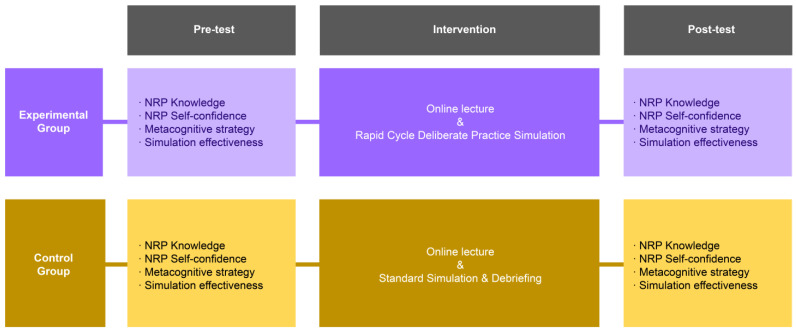
Research design of this study.

**Table 1 healthcare-12-00104-t001:** Homogeneity test of participant characteristics and dependent variables (*N* = 57).

Characteristics	Categories	Experimental Group (*n* = 29)	Control Group (*n* = 28)	χ^2^ or *t*	*p*
		*n* (%)	*n* (%)		
Sex	Male	9 (31.0)	9 (32.1)	0.01	0.928
	Female	20 (69.0)	19 (67.9)		
Major satisfaction	Dissatisfied	0 (0.0)	2 (7.1)	2.37	0.305
	Neutral	13 (44.8)	10 (35.7)		
	Very satisfied	16 (55.2)	16 (57.1)		
Satisfaction with clinical practice	Neutral	9 (31.0)	13 (46.4)	1.42	0.233
	Very satisfied	20 (69.0)	15 (53.6)		
Satisfaction with simulation training	Neutral	11 (37.9)	18 (64.3)	3.96	0.057
	Very satisfied	18 (62.1)	10 (35.7)		
Demand for simulation training for nursing major	Yes	28 (96.6)	27 (96.4)		0.746 *
	No	1 (3.4)	1 (3.6)		
NRP knowledge, M (SD)		9.59 (3.61)	8.25 (4.01)	1.32	0.191
NRP self-confidence, M (SD)		50.17 (9.87)	47.46 (13.96)	0.02	0.400
Clinical decision-making skill, M (SD)		147.86 (16.45)	141.07 (12.72)	1.72	0.091
Simulation effectiveness, M (SD)		44.10 (7.24)	42.21 (7.91)	0.94	0.351

* Fisher’s exact test; M: Mean; SD: Standard Deviation; NRP: Neonatal resuscitation program. In the preliminary homogeneity test of the experimental and control groups, the two groups showed no significant differences in their general characteristics and dependent variables.

**Table 2 healthcare-12-00104-t002:** Comparison of dependent variables between the groups at pre- and post-test (*N* = 57).

Variables	Groups	Pre-Test	Post-Test		Pre-Post Difference	
		M (SD)	M (SD)	t (*p*)	M (SD)	t (*p*)
NRP knowledge	Experimental group	9.59 (3.61)	14.24 (3.20)	2.25 (0.028)	4.66 (4.37)	0.38 (0.703)
	Control group	8.25 (4.01)	12.46 (2.73)		4.21 (4.31)	
NRP self-confidence	Experimental group	50.17 (9.87)	65.66 (7.78)	4.67 (<0.001)	17.55 (7.86)	1.45 (0.045)
	Control group	47.46 (13.96)	57.86 (6.70)		10.39 (17.15)	
Clinical decision-making skill	Experimental group	147.86 (16.45)	156.86 (12.37)	6.03 (<0.001)	9.00 (12.74)	3.13 (0.003)
	Control group	141.07 (12.72)	137.57 (11.77)		−3.67 (17.39)	
Simulation effectiveness	Experimental group	44.10 (7.24)	52.10 (5.49)	3.51 (0.001)	8.00 (6.46)	1.99 (0.042)
	Control group	42.21 (7.91)	46.61 (6.33)		4.39 (7.25)	

Experimental group (*n* = 29); Control group (*n* = 28); M: Mean; SD: Standard Deviation.

## Data Availability

The data presented in this study are available upon request from the corresponding author. The data are not publicly available due to lack of permission from study subjects to access the data.

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
