# Peer review of "Development and Effectiveness of a Rapid Cycle Deliberate Practice Neonatal Resuscitation Simulation Program: A Quasi-Experimental Study"

_healthcare, 2024, doi:10.3390/healthcare12010104_

Round 1

Reviewer 1 Report

Comments and Suggestions for Authors

This is a very relevant study in improving the skills of novice learners of NRP. Study design was adequate to detect a difference in pre and post intervention improvement in knowledge, decision making skills and confidence of the learners. A description of the script used by the instructors to provide RCDP feedback to the intervention group would be useful to the readers. Providing more information on the debriefing method used for the control group would give the readers a comparison of the different methodologies.  Please provide the validated research tools used to measure the data presented in the study as supplemental material. 

Comments on the Quality of English Language

Minor grammatical errors need to be corrected. 

Reviewer 2 Report

Comments and Suggestions for Authors

I congratulate the authors in their attempt to improve the quality of neonatal resuscitation through exploration of different simulation techniques. Rapid Cycle Deliberate Practice has been utilized in many different areas of medicine, but has not yet been incorporated in the NRP educational program. As such, this study has the potential to help provide more diverse simulation options to help maximize the NRP learning experience. 

My suggestions / edits are minor and they are as follows:

Abstract: 

Line 11: The Rapid Cycle Deliberate Practice neonatal resuscitation program (NRP) does not exist. Instead of applying this title to the NRP, I suggest either adding the word simulation before program as was done in the title (as in neonatal resuscitation simulation program as well as getting rid of (NRP)).  The other option would be to re-word to .... The Rapid Cycle Deliberate Practice for neonatal resuscitation.

Line 17: Once again, suggest replacing .... RCDP NRP with for example,  .... RCDP simulation during NRP training.....

Line 21: Once again, suggest changing to..... we found that using RCDP simulation during NRP training......

Introduction:

Line 50: Suggest changing repeatable to repetitive 

Methods:

Line 120: What do you mean by insincere response? How would you know?

Line 124: a NICU nurse instead of an NICU nurse

Line 242: What is meant by XXXX after the word questionnaire?

Reviewer 3 Report

Comments and Suggestions for Authors

Manuscript by Yang et al study benefits of Rapid Cycle Deliberate Practice compared to traditional NRP in nursing students. Manuscript is interesting though authors may want to address some concerns

1. The study was carried out with nursing students. How many of them had background knowledge of NRP?

2. Were these students exposed to any real life scenarios?

3. Was it NRP essentials course or advanced course? 

4. Were these students expected to lead extensive resuscitation?

5. How were students scored when one of the team members was not performing well?

6. Please explain figure 1 in more details. What comprised of blocks A, B, C and D?

7. How much time was between the training and post test?

8. How did authors check for periodic retention of skills?

9. Please mention with IRB approval and written consent from participants. 

10. Was this voluntary activity of students? Were they paid for the participation?
